# An ABA Functional Analogue B2 Enhanced Salt Tolerance by Inducing the Root Elongation and Reducing Peroxidation Damage in Maize Seedlings

**DOI:** 10.3390/ijms222312986

**Published:** 2021-11-30

**Authors:** Shiying Geng, Zhaobin Ren, Lijun Liang, Yumei Zhang, Zhaohu Li, Yuyi Zhou, Liusheng Duan

**Affiliations:** 1State Key Laboratory of Plant Physiology and Biochemistry, Engineering Research Center of Plant Growth Regulator, Ministry of Education & College of Agronomy and Biotechnology, China Agricultural University, No. 2 Yuanmingyuan Xi Lu, Haidian District, Beijing 100193, China; gengshiying@cau.edu.cn (S.G.); madfrog@cau.edu.cn (Z.R.); lianglijun0620@163.com (L.L.); lizhaohu@cau.edu.cn (Z.L.); 2College of Agronomy, Qingdao Agricultural University, Qingdao 266109, China; zhangcui2003@163.com

**Keywords:** plant growth regulators, B2, maize, salt tolerance

## Abstract

Salt stress negatively affects maize growth and yield. Application of plant growth regulator is an effective way to improve crop salt tolerance, therefore reducing yield loss by salt stress. Here, we used a novel plant growth regulator B2, which is a functional analogue of ABA. With the aim to determine whether B2 alleviates salt stress on maize, we studied its function under hydroponic conditions. When the second leaf was fully developed, it was pretreated with 100 µM ABA, 0.01 µM B2, 0.1 µM B2, and 1 µM B2, independently. After 5 days treatment, NaCl was added into the nutrient solution for salt stress. Our results showed that B2 could enhance salt tolerance in maize, especially when the concentration was 1.0 µMol·L^−1^. Exogenous application of B2 significantly enhanced root growth, and the root/shoot ratio increased by 7.6% after 6 days treatment under salt stress. Compared with control, the ABA level also decreased by 31% after 6 days, which might have resulted in the root development. What is more, B2 maintained higher photosynthetic capacity in maize leaves under salt stress conditions and increased the activity of antioxidant enzymes and decreased the generation rate of reactive oxygen species by 16.48%. On the other hand, B2 can enhance its water absorption ability by increasing the expression of aquaporin genes *ZmPIP1-1* and *ZmPIP1-5*. In conclusion, the novel plant growth regulator B2 can effectively improve the salt tolerance in maize.

## 1. Introduction

Salinity is a major threat in the modern agriculture, inhibiting growth and reducing yield, consequently leading to economic loss for the farming organization and the country [1,2]. In general, high salt concentrations have adverse effects on plant metabolic processes, including nutritional imbalance, oxidative stress, water stress, membrane disturbance, and ionic toxicity [3,4,5,6]. When plants are exposed to salt stress, specific signals are generated to initiate their response to salt stress, thus improving their tolerance to salt stress [7,8]. In order to solve the problem of salinity, people have found many methods including improving the physical and chemical properties of land, cultivating new crop varieties that are salt-tolerant, and using plant hormones or plant growth regulators. The plant growth regulators play a vital role in improving salt tolerance of plants.

As is well known, hormonal regulation is very important in stress response [9,10,11]. Abscisic acid (ABA), a plant stress hormone, can amplify primary signal and stimulate other signaling networks [12,13]. ABA can regulate the response of root growth to phosphorus starvation in barley, and this effect is related to the concentration of cytokinin and auxin [14]. ABA could regulate spikelet abortion of barley and affect yield potential of barley under salt stress [15]. The ABA signaling cascade was found to enhance reactive oxygen species (ROS) level functions as an intermediary signaling component in stomatal closure, which could counteract the effects of water-deficit and ionic unbalance [16,17]. Salt stress results in osmotic stress in plants, which weakens respiration and photosynthesis; meanwhile, a large number of ROS are produced. ROS promotes the biosynthesis of ABA, activating downstream transcription factors (TFs), which can induce salt-responsive genes, thereby improving plant salt tolerance [18,19]. Although ion toxicity is detoxified through the ROS pathway, its downstream is the same as that of hormonal pathways. Therefore, ABA can enhance the role of the ROS pathway by enhancing its same downstream network of ROS pathway. The genes of relating to defensive enzymes such as POD, SOD, and CAT were induced by increased ABA and ROS is scavenged [20]. These studies indicated that ABA-associated pathway plays an important role in salt stress response.

The application of exogenous ABA can improve the salt tolerance of crops, but the chemical properties of ABA are unstable and easy to decompose, and the cost of artificial synthesis of natural ABA is high, which limits its application in agricultural production [21]. Pyrabactin is the first artificial ABA analog that specifically binds to a subset of PYL receptors in *Arabidopsis thaliana* and inhibits seeds germination [22]. AM1 (ABA mimic 1), also known as quinabactin, is a more effective pan-agonist of PYLs, one that can enhance drought resistance by reducing water loss and activating stress-responsive genes. Therefore, the analogs of ABA have an important role in increasing salt tolerance and other stress tolerance [23,24].

We adopted a subactive structure splicing method, integrating ABA, pyrabactin, and coronatine, then synthesized compound B2 [8,25,26,27]. B2 enhances salt tolerance and drought resistance of wheat [25,27]. To determine whether B2 has similar effects to ABA on salt tolerance in maize, we used Zhengdan 958, which is sensitive to salt stress and widely cultivated in China, to study the effect of B2 in enhancing salt tolerance on the basis of the physiological changes and different expression level of related genes.

## 2. Results

### 2.1. B2 Improved Seed Germination under Salt Stress Conditions

One of the important characteristics of maize with high stress tolerance is seed germination rate. ABA can inhibit seed germination. After the maize seeds were treated with 100 µM ABA, the germination rate was significantly reduced by 59.1% under normal conditions but was significantly increased under salt stress conditions. Compared with ABA, B2 showed an interesting effect on seeds germination. The germination rate of seeds treated with different concentration B2 did not show a decrease trend under normal conditions but increased significantly under salt stress by more than 74.2%. Compared with ABA, the germination rate of seeds treated with B2 decreased 6.27% under salt stress conditions, but this difference was not significant (Figure 1).

### 2.2. B2 Improved the Biomass of Maize Seedlings under Salt Stress

The morphology of maize seedlings with different treatments were remarkable according to the diagram taken after 6 days of treatment (Figure 2). Under normal conditions, maize seedlings treated with B2 was similar to the control plants, but better than the plants treated with 100 µM ABA. Maize roots with B2 treatment were significantly induced after 6 days, and the effect of 1.0 µM B2 was the best (Figure 2b). Under salt conditions, the growth of maize seedlings was inhibited, while the seedlings treated with 1.0 µM B2 and 100.0 µM ABA grew better than control plants. What is more, B2 was able to promote maize root elongation under salt stress conditions, especially 1.0 µM B2. Compared with ABA, B2 showed a similar effect in root elongation (Figure 2d).

Statistical analysis also revealed that there was a difference in maize seedlings among different treatments under salt stress conditions. The data of growth status showed that the effect of 1.0 µM B2 was similar with ABA. Under normal conditions, the plant height of seedlings treated with 1.0 µM B2 was significantly increased by 24.9% compared to the control plants (Figure 3a). In addition, the dry weight of seedlings was significantly increased by 26.72% compared to control plants (Figure 3b). Under stress conditions, the plant height and dry weight of 1.0 µM B2 treatment was better than control plants, and the plant height and shoot dry weight were significantly increased by 26.25% and 38.19% compared to the control (Figure 3).

Moreover, the roots showed a significant difference among different treatments in morphology. The dry weight of maize root with 0.01 µM and 0.1 µM treatments had no significant difference compared to control plants, but the root with 1.0 µM B2 treatment significantly increased by 49.42% compared to the control plants under stress conditions (Figure 4a). What is more, we not only observed a 56.33% increase of maize total root length and 22.45% increase of root surface area pretreated with 1.0 µM B2 under salt stress conditions, but also observed 11.09% and 14.08% increases, respectively, under normal conditions (Figure 4b,c). Exogenous application of 1.0 µM B2 also significantly increased the root/shoot ratio by 29.94% under normal conditions (Figure 4d).

### 2.3. The Endogenous Hormone ABA Level Changed with B2 Treatment

The plant hormones play an important role as messengers for stress signaling, and therefore we measured the major endogenous hormone ABA content. It can be seen from the above results that the functional effect of 1.0 µM B2 was closer to ABA, and therefore we chose the best concentration of B2 to study further. When maize seedlings grew under the normal condition, the results showed that the ABA level of seedlings with ABA and B2 pretreatment decreased sharply by 30.95% and 31.16%, respectively, after 6 days (Figure 5). This suggests that after 6 days, the inhibition effect of ABA was removed, resulting in root elongation. Under salt conditions, the content of ABA in maize seedlings with ABA and B2 treatment also decreased sharply by 33.18% and 46.75%, respectively, after 12 days (Figure 5).

### 2.4. B2 Improved Maize Morphology by Enhancing Photosynthetic under Salt Stress

The biomass of maize seedlings was decreased under salt stress, and it was inseparable from the synthesis of organic matter. In the presence of exogenous ABA and B2, photosynthetic rate decreased sharply by 3.42 and 3.13 μmol/m^2^/s without salt stress, respectively (Figure 6a). Compared with the control plants, the photosynthesis rate in seedlings decreased dramatically under salt stress. However, those treated with 1.0 µM B2 under salt stress maintained a much higher photosynthetic rate in leaves than control plants and ABA-treated plants, which increased sharply by 20.06% and 17.65%, respectively. The concentration of intercellular CO_2_ and the content of chlorophyll showed a remarkable interaction between B2 and salt, increasing significantly by 48.53% and 15.83%, respectively (Figure 6b,e). The rate of transpiration decreased sharply under salt stress but increased sharply by 22.70% after ABA pre-treatment (Figure 6c). ABA pre-treatment resulted in a significant decrease in stomatal conductance without salt stress, but B2 had no significant effect in stomatal conductance. In the pre-treatment of exogenous ABA and B2, stomatal conductance increased dramatically by 67.00% and 66.02% under salt stress, respectively (Figure 6d). In addition, the chlorophyll fluorescence showed significant difference in all treatments after 0–24 h (Figure 6f). After 48 h salt stress, chlorophyll fluorescence decreased by 0.049 (Fv/Fm), and ABA and B2 treatment showed no significant difference. After 72 h pre-treatment with exogenous ABA and B2, stomatal conductance increased remarkably by 6.83% and 7.20%, respectively, under salt stress. These results indicated that the photosynthesis of maize was increased with B2 treatment, and B2 has similar function to ABA in maintaining photosynthesis.

### 2.5. B2 Reduced the Damage of Maize Leaves under Salt Stress

Salt-induced stress increased relative electrolyte leakage significantly, suggesting high electrolyte leakage, but treatment with ABA and B2 decreased the leakage by 22.42% and 19.06%, respectively, under salt stress (Figure 7); without salt stress, the relative electrolyte leakage was little affected by ABA and B2. The soluble protein content of maize seedlings increased under salt stress, and it also significantly increased by 8.42% in the seedlings treated with 1.0 µM B2 after 3 days. Under normal conditions, the soluble protein content of maize seedlings treated with 100.0 µM ABA and 1.0 µM B2 significantly increased by 19.26% and 20.71%, respectively. This suggests that B2 acclimation to salt was observed in electrolyte leakage and soluble protein and was accompanied by a reduction in cellular damage in response to salt stress.

### 2.6. B2 Reduced the Degree of Peroxide in Maize Leaves under Salt Stress

The excessive ROS can increase the degree of peroxide, and it will damage plant cells. B2 did not affect the production of free radicals (O^2−^) in leaves under normal conditions. However, the O^2−^ production in leaves was increased under salt stress, although treatment with ABA and B2 decreased its level (Figure 8a). Pre-treatment with B2 decreased the leakage considerably by 29.68% after 12 days of salt stress. Meanwhile, the antioxidant enzymes were analyzed after salt stress. Activities of antioxidant enzymes (SOD, CAT, and POD) were remarkably affected by salt stress. However, plants pre-treated with B2 had sharply increased SOD activity by 16.89% and 25.28%, respectively, after 6 days and 12 days of salt stress (Figure 8b). The activity of CAT in plants pre-treated with ABA and B2 increased sharply by 13.07% and 11.57% after 12 days of salt stress, respectively (Figure 8c). The activity of POD was also significantly increased by 53.34% at 12 days after salt stress (Figure 8d). These results indicated that B2 has similar function to ABA and suggest that B2 increased the activity of antioxidant enzymes under salt stress.

### 2.7. B2 Can Enhance the Bibulous Capacity by Increasing the Expression of Aquaporins

Under salt stress conditions, the expression levels of *ZmAO* in maize seedlings treated with 1.0 µM B2 decreased by 14.33% and 54.37% compared to control plants after 3 h and 6 h, respectively (Figure 9a), and the expression levels of *ZmVp14-2* was reduced by 95.94% after salt stress for 3 h (Figure 9b). Our results show that B2 can increase the expression levels of *ZmPIP1-1* after 24 h and 48 h under salt stress conditions, whose expression level was increased by 172.65% and 49.55%, respectively (Figure 9c). The expression levels of *ZmPIP1-5* were increased by 155.07% after 24 h under salt stress (Figure 9d). The results indicated that B2 inhibited ABA biosynthetic genes expression and increased the expression of aquaporins.

## 3. Discussion

Salinized soils are a major threat to global food security [28,29]. According to the FAO (2008), at least 800 million hectares of land around the world is currently salinized. The main sensitive period of plants to salt is the seedling stage. ABA is an important plant hormone that plays an important role in salt stress, including stomatal conductance, cell signal transduction, and plant growth regulation. This study investigated the effects of ABA functional analogue B2 on growth, photosynthesis, antioxidant enzyme activity, reactive oxygen species, and related genes of maize seedlings under salt stress. Plant height and dry weight of seedlings are common characteristics to determine plant growth status. Previous studies proposed the use of plant height and dry weight as screening tests for salt tolerance under salt stress [30,31]. In this study, we found that B2 significantly increased plant height and dry weight of maize (Figure 3). Further studies showed that B2 increased root length, root surface area, and root dry weight under salt stress (Figure 4), thus enhancing the salt tolerance of maize seedlings.

The main reason why maize can grow normally under salt stress is that photosynthesis can be maintained [32,33]. Under salt stress, B2 significantly increased photosynthetic rate, intercellular CO_2_ concentration, and chlorophyll content (Figure 6), and decreased electrical conductivity (Figure 7). In addition, B2 can maintain chlorophyll activity (Figure 6f). Therefore, B2 enhanced the salt tolerance of maize seedlings. ROS can increase the level of peroxide. Plants improve their resistance to salt stress by regulating the activity of their antioxidant system, and therefore the antioxidant enzymes CAT, SOD, and POD, which are usually used to show antioxidant capacity, increase [34,35,36]. When plants are subjected to oxidative stress, the plant’s antioxidant defense begins. O^2−^ is converted to H_2_O_2_ by SOD in cytoplasm, peroxisomes, chloroplasts, eoplasts, and mitochondria [37], which is then detoxicated by catalase and peroxidase [38,39,40]. POD activity was enhanced by ABA mimic (AM1). It has been reported that exogenous ABA can significantly increase CAT activity, POD activity, SOD activity, and GR activity. It increased the content of ascorbic acid and decreased the content of carotenoid, a-tocopherol, and glutathione. In our study, under salt stress, the activities of SOD and CAT in plant leaves were significantly increased after B2 treatment for 6 days, and the formation rate of O^2−^ was significantly decreased (Figure 8). In conclusion, B2 can improve the ROS scavenging ability of maize by increasing the activity of antioxidant enzymes in maize seedlings under salt stress. Thus, oxidative damage is reduced by B2. It indicated that the salt tolerance of maize was improved.

The results showed that the expression levels of *ZmPIP1-1*, *ZmPIP1-5*, *ZmVP14-2*, and *ZmAO* genes in maize were significantly changed under abiotic stress. When *ZmPIP* expression is upregulated, salt tolerance is increased [41,42]. In our study, we found that the expression level of *ZmPIP1-1* at 24 h was higher than that of the control, and the expression level of *ZmPIP1-5* was higher than that of the control at 24 h and 48 h (Figure 9). Studies have shown that ABA biosynthesis genes *ZmVP14-2* and *ZmAO* are highly expressed under salt stress, and their salt tolerance increases with the increase of ABA content [42] (Zhang et al., 2016). In our study, the expression of ABA biosynthetic genes *ZmVP14-2* and *ZmAO* were downregulated or remained unchanged after B2 treatment under salt stress (Figure 9). This result differs from previous studies. It may be that B2 reduces salt stress injury. Maize does not require more ABA, promoting salt stress-induced defense genes.

In conclusion, ABA functional analogue B2 can enhance the salt tolerance of maize by enhancing the activity of antioxidant enzymes and the scavenging ability of reactive oxygen species. The leaf photosynthesis was significantly enhanced, which could make maize grow close to normal under salt stress. In addition, B2 has a unique regulatory mechanism, just as it significantly reduces ABA content in maize under salt stress. The results of this study provided an economical method for solving maize salt stress.

## 4. Materials and Methods

### 4.1. The Synthesis of B2

The synthesis of B2 was according to the method that reported by Zhou et al. [25,26]. Firstly, we obtained substituted benzoyl chloride by replacing benzoic acid with dichlorosulfoxide, and then B2 by replacing benzoyl chloride with cyclopropyl amino acid.

### 4.2. Plant Material and Treatment

The cultivar, named “Zhengdan 958” was used in this study. It is sensitive to salt stress and widely cultivated in China. The seeds were collected from China Agricultural University Sinong Seeds Company.

In the experiments, the seedlings were equally divided into groups and cultured hydroponic culture in basins containing nutrient solution, which contains 0.05 mM KNO_3_, 1 mM CaCl_2_, 0.5 mM MgSO_4_, 0.1 mM KH_2_PO_4_, 0.1 mM NaFeEDTA, 0.03 mM H_3_BO_3_, 0.005 mM MnSO_4_, 0.00003 mM (NH_4_)_2_MoO_4_, 0.0025 mM ZnSO_4_, and 0.008 mM CuSO_4_. The growth environment of maize was set at 16 h/8 h photoperiod, 28 °C/25 °C day/night temperature, 50% relative humidity, and 400 µmol/m^2^ per photon flux density.

When the second true leaves were fully developed, 100 µM ABA, 0.01 µM B2, 0.1 µM B2, and 1.0 µM B2 were sprayed, respectively. After 5 days of treatment with ABA or B2, NaCl was added to the nutrient solution. The concentration of NaCl was 100 mM in the nutrient solution, which was used to simulate salt stress. The phenotype, the height of plants, and the photosynthetic targets were measured after 6 days of treatment. After 6 days of treatment with NaCl, the length and surface area of root were measured by using the EPSON EXPRESSIONTM 1680 root scanner and WinRHIZO software. We then washed the corn seedlings and placed them in envelopes and dried them at 120 °C for 1 h, which was followed by 80 °C until the weight remained unchanged. Finally, the dry weights of root and shoot were measured by scale. Specimens were collected at 3, 6, and 12 days after salt treatment. Three biological replicates were collected for each treatment, and three plants were collected for each sample. Then, it was quickly frozen with liquid nitrogen and immediately stored in a −80 °C refrigerator.

### 4.3. Electrolyte Leakage Measurement

Osmotic system data are among the most important results reflecting whether plants grow under stress. In order to study the osmotic system of maize under stress, we measured the electrical leakage of maize leaves. The electrolyte leakage was measured by evenly cutting the newly expanded maize leaves into small pieces, weighing 0.2 g tissue samples into a 15 mL centrifuge tube, adding 10 mL deionized water at 12,000 rpm/min, centrifuging at room temperature for 24 h, and measuring values (L0) with an EC 215 Hanna instrument conductivity meter. Then, we boiled the water bath for 30 min, cooled it to room temperature, and measured the conductivity again (Lt). Three replicates were set for each treatment. The relative electrolyte leakage was calculated as follows:Relative electrolyte leakage (%) = Lt × 100/L0

### 4.4. Soluble Protein

The method of measuring soluble protein content is Coomassie blue staining. Samples were collected after 3, 6, and 12 days of treatment. A total of 5 mL Coomassie brilliant blue G-250 was add into the 1 mL measuring liquid of sample. The soluble protein content was determined by a spectrophotometer by colorimetric determination at 595 nm after mixing the mixture for 2 min. Protein content was calculated according to standard curve. Thus, the protein content was calculated as follows:Soluble protein content (mg/g FW) = (C × V/a)/W

C is protein content, V is the total volume of extract, a is the volume of measuring extract, and W is the weight of the sample.

### 4.5. Photosynthesis and Chlorophyll Fluorescence

The photosynthetic parameters of leaf surface were measured by portable photosynthetic instrument LI-6400 (LI-Cor Inc., Lincoln, NE, USA). The light intensity was 1000 µmol/m^2^/s, and the flow rate was 500 mL/s when the parameters were measured. In addition, the temperature was 25–28 °C. We selected the third leaf for experiment, and three biological replicates were collected for each treatment.

Chlorophyll fluorescence was measured by the portable fluorescence instrument German WALZ PAM2100 after 0, 24, 48, and 72 h of treatment. The time of dark processing was 20–30 min. Three biological replicates were collected.

The chlorophyll content was measured by select the leaf of latest expanded for the experiment. We collected leaves of 0.1 g and cut them into filaments. We then put the filaments of the leaf into 25 mL acetone solution (80%) and soaked the sample for 24 h in the dark. OD was measured by spectrophotometer by colorimetric determination at 663 nm and 645 nm. The chlorophyll content was calculated as follows:Ca = (12.7 × D663 − 2.69 × D645) × V/1000 W
Cb = (22.9 × D645 − 4.68 × D663) × V/1000 W

Ca and Cb, respectively, are concentrations of chlorophyll a and chlorophyll b. D663 and D645 are absorbance of 663 nm and 645 nm, respectively. W is fresh weight of sample, and V is volume of extract.

### 4.6. Superoxide Radical Estimation

The O^2−^ was measured by monitoring nitrite formation from hydroxylamine following the method of Xie et al. [43]. The production of H_2_O_2_ was measured following the method described in Verma and Mishra [44]. Since the effect of 1.0 μM B2 was closer to ABA, 1.0 μM B2 was used here. Firstly, we grinded 0.5 g of leaves of the plant into the mortar, and then put them into a 10 mL centrifuge tube after putting 2 mL phosphate buffer (50 mM, pH 7.0) into the mortar. After this, we washed the mortar with the 3 mL extract and put 3 mL extract into the centrifuge tube. We then centrifuged the mixture at 12,000 rpm for 25 min at 4 °C, and then we placed it on ice for 10 min. The supernatant was moved to new tube at 4 °C. We then put 0.5 mL phosphate buffer (50 mM, pH 7.8) and 1 mL hydroxylamine hydrochloride (1 mM) into the tube at 25 °C for 1 h. After that, we put 1 mL p-aminobenzene sulfonic acid (17 mM) and 1 mL alpha-naphthylamine (7 mM) into the tube and mixed them well. OD was measured by spectrophotometer by colorimetric determination at 530 nm after place at 25 °C for 20 min. According to the measured OD_530_, we checked the standard curve of N•O^2−^ to obtain the content of NO^2−^. The [O_2_] was twice the [NO^2−^], and this value is M. The t is the time of sample reacting with the hydroxylamine, and m is protein content in the sample. The generation rate of •O^2−^ was calculated as follows:Generation rate of •O^2−^ (nmol·min^−1^·mg^−1^) = M/(t × m)

### 4.7. Enzyme Extraction

After 3, 6, and 12 days of treatment, the youngest fully expanded leaves from five plants in each replication were sampled for enzyme extraction and determination. The method for determination of the activities of SOD, CAT, and POD was that of Xie et al. The samples of determination of SOD, CAT, and POD were collected at the concentration of 1.0 µM B2.

### 4.8. Determination of Endogenous Hormone Content

We extracted endogenous hormones and made standard curves. The liquid phase conditions included 254 nm wavelength, 0.8 mL/min flow rate, and the mobile phase. The mobile phase included methanol and 0.5% acetic acid, and their ratio was 55:45 at 0–4 min, 45:55 at 4–8 min, and 55:45 at 8–10 min. The linear equations of ABA were y = 0.0159x − 2.0484, R^2^ = 0.9971; the linear equation of y = 0.0595x − 3.5995, R^2^ = 0.9979; and the linear equation of y = 0.0697x − 3.4424, R^2^ = 0.997.

### 4.9. Gene Expression

The samples were collected at 3, 6, 12, 24, 48, and 72 h after treatment. According to the experiment manual of the manufacturer, total RNAs were isolated with a Trizol reagent (TransGen, Beijing, China), and each sample was in three biological replicates.

Relative expression levels of DEGs were quantified by quantitative real-time PCR (qPCR) using the maize actin gene as internal control. Using Takarade SYBR ^®^ Premix Ex Taq Mix (Perfect Real Time) Kit, qPCR reactions were performed on an ABI-7500 Fast PCR System, 95 °C for 30 s, 40 cycles at 95 °C for 5 s, and 60 °C for 34 s. The 15 μL reaction mixture contained 1.5 μL cDNA, 7.5 μL 2 × SYBR ^®^ Premix Ex Taq TM, 0.3 μL of each primer, 0.3 μL 50 × ROX Reference DyeII, and 5.1 μL water. The relative expression level was calculated with the 2-ΔΔCt method. The primers used for qPCR are listed in Appendix A.

### 4.10. Statistical Analysis

Data shown are the mean ± standard deviation of three independent experiments. Mean differences were compared using the statistical software data processing system (SPSS 17.0), followed by the multiple comparisons, and the differences between group means were considered significant at *p* < 0.05.

## Figures and Tables

**Figure 1 ijms-22-12986-f001:**
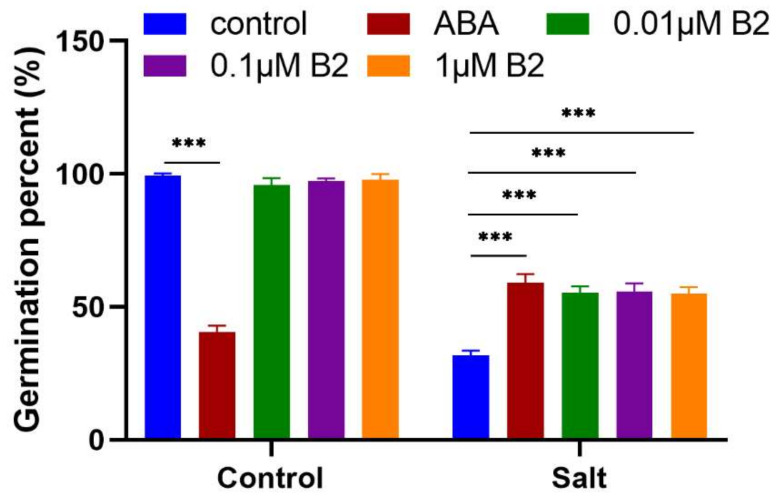
The germination percentage of maize treated with different concentration of B2 was improved under the salt stress. ABA concentration was 100 Μm. Values were significantly different from controls by *t*-test at *** *p* ≤ 0.001. Bars mean SD (*n* = 25).

**Figure 2 ijms-22-12986-f002:**
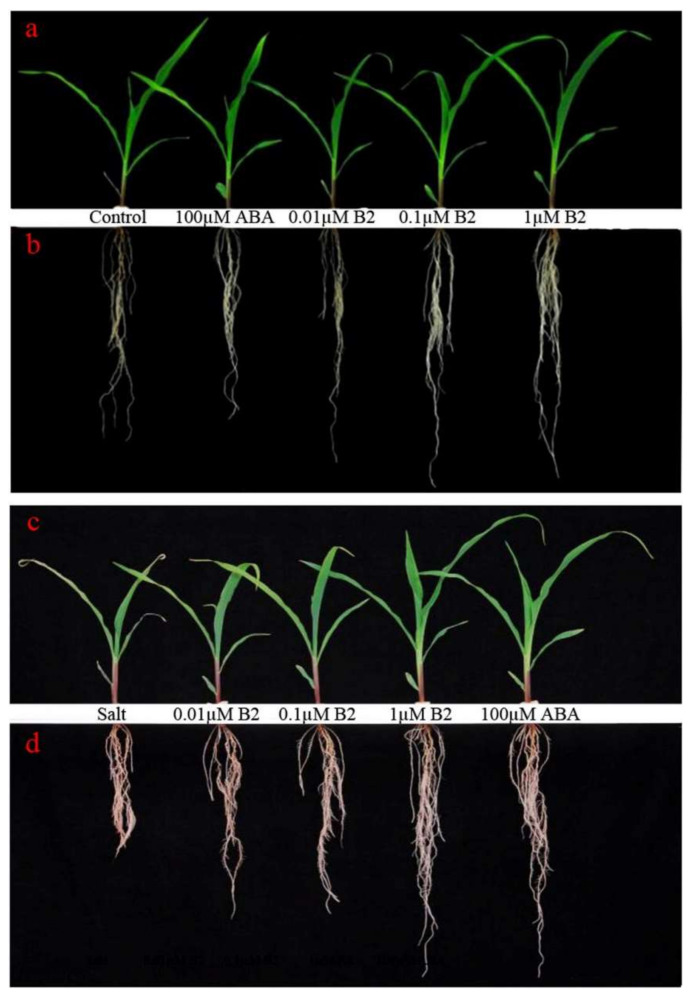
The morphology of maize seedlings. (**a**,**b**) The maize seedlings in normal solution treated with 100 μM ABA or 0.01 μM, 0.1 μM, or 1 μM B2 after 6 days. (**c**,**d**) The maize seedlings under salt stress conditions treated with ABA or B2 after 6 days. The seedlings are arranged from small to large from left to right.

**Figure 3 ijms-22-12986-f003:**
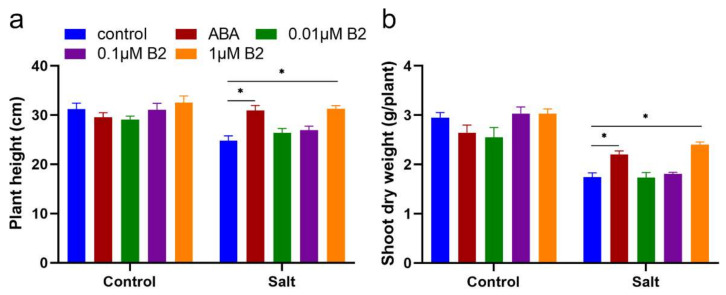
Plant height and dry weight of maize seedlings. (**a**) Height of maize shoot. Height of maize shoot is indicated on the *y*-axis. (**b**) Dry weight of maize shoot. This was measured after 6 days under normal conditions and salt stress conditions. ABA concentration was 100 μM, B2 concentrations were 0.01 μM, 0.1 μM, and 1 μM. Dry weight of shoot is indicated on the *y*-axis. Values were significantly different from controls by *t*-test at * *p* < 0.05. Bars mean SD (*n* = 9).

**Figure 4 ijms-22-12986-f004:**
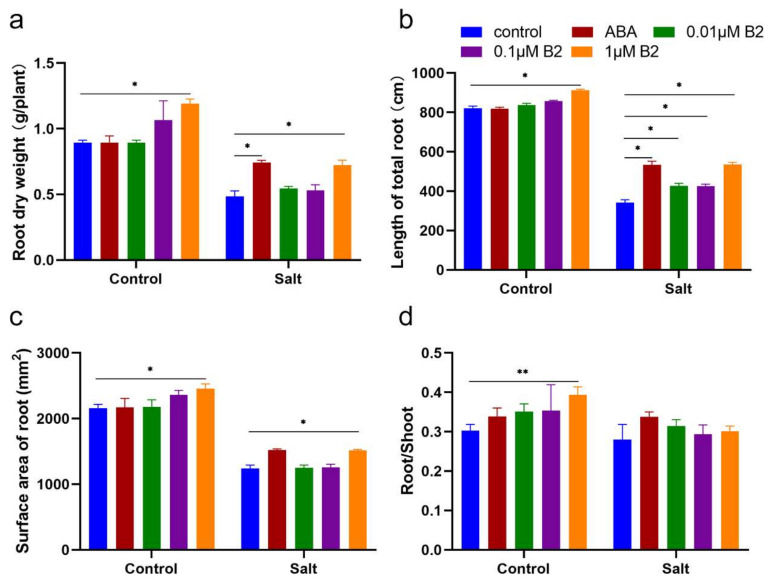
Maize root growth. (**a**) The dry weight of roots. (**b**) The length of total roots. (**c**) The surface area of roots. (**d**) The root/shoot ratio. These measurements were taken 6 days after different treatment. ABA concentration was 100 μM, B2 concentrations were 0.01 μM, 0.1 μM, and 1 μM. Values were significantly different from controls by *t*-test at * *p* < 0.05 or ** *p* ≤ 0.01. Bars mean SD (*n* = 9).

**Figure 5 ijms-22-12986-f005:**
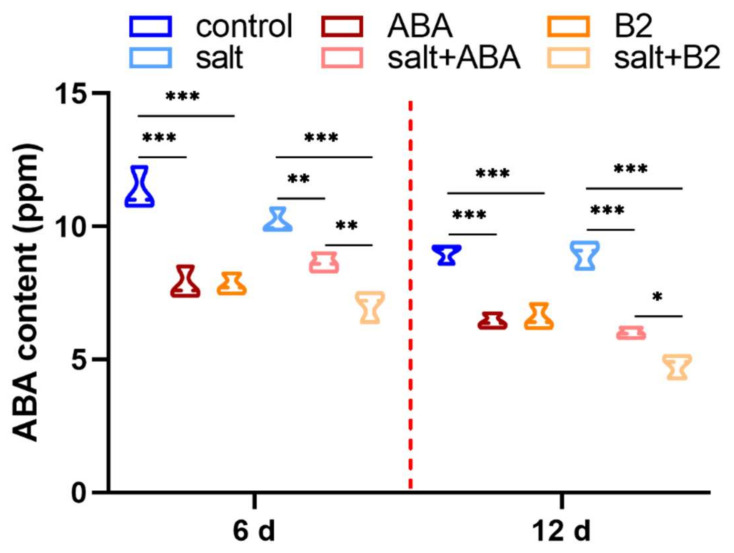
Content of endogenous ABA. ABA concentration was 100 μM, and the concentration of B2 was 1.0 µM (the functional effect of 1.0 µM B2 was closer to ABA, and therefore we chose the concentration of B2 to be 1.0 µM). The measurements were taken 6 days and 12 days after salt stress. Content (ppm) is indicated on the *y*-axis. Time is indicated on the *x*-axis. Values were found to be significantly different from controls by *t*-test at * *p* < 0.05, ** *p* ≤ 0.01, or *** *p* ≤ 0.001. Bars mean SD (*n* = 9).

**Figure 6 ijms-22-12986-f006:**
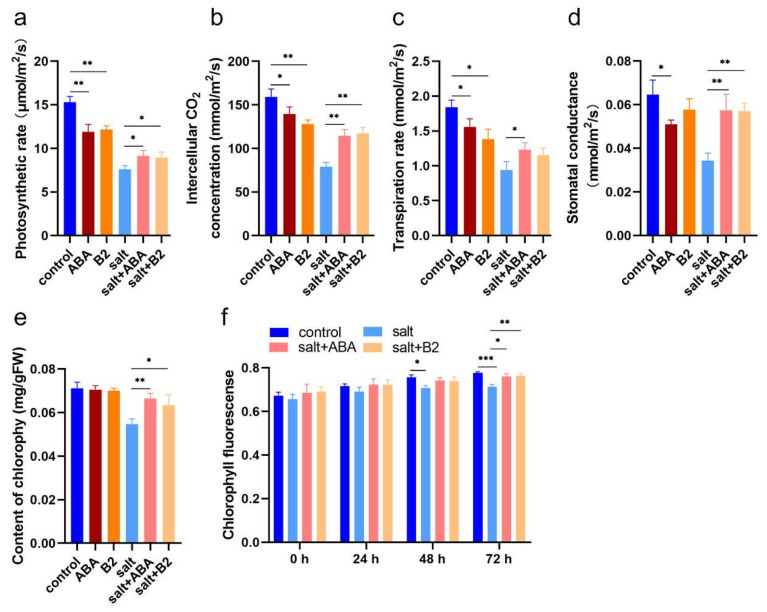
Photosynthesis and chlorophyll fluorescence of maize. (**a**) Photosynthetic rate of different treatments. (**b**) Intercellular CO_2_ concentration of different treatments. (**c**) Transpiration rate of different treatments. (**d**) Stomatal conductance of different treatments. (**e**) The content of chlorophyll of different treatments. (**a**–**e**) The latest unfolding leaf measured after 6 days of salt treatment. (**f**) The chlorophyll fluorescence measured after 0, 24, 48, and 72 h of salt treatment. The concentration of B2 was 1.0 µM, and ABA concentration was 100 μM. Values were significantly different from controls by *t*-test at * *p* < 0.05, ** *p* ≤ 0.01, or *** *p* ≤ 0.001. Bars mean SD (*n* = 9).

**Figure 7 ijms-22-12986-f007:**
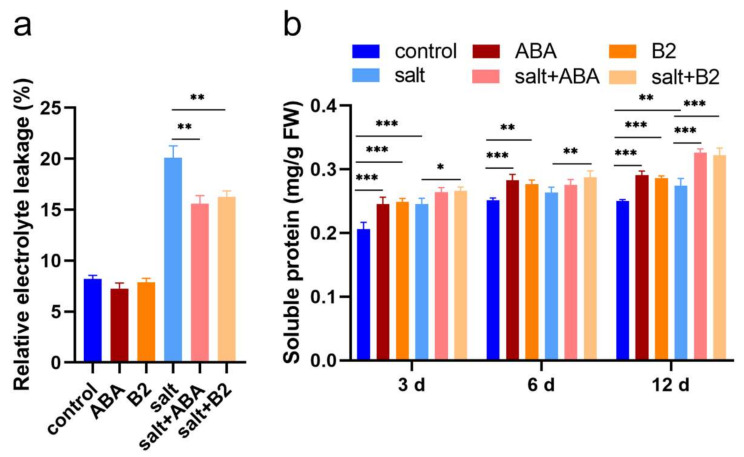
Electrolyte leakage and soluble protein of maize seedlings. (**a**) Electrolyte leakage was measured after 6 days salt stress. (**b**) Soluble protein was measured after 3, 6, and 12 days of salt stress. The concentration of B2 was 1.0 µM, and ABA concentration was 100 μM. Values were significantly different from controls by *t*-test at * *p* < 0.05, ** *p* ≤ 0.01, or *** *p* ≤ 0.001. Bars mean SD (*n* = 9).

**Figure 8 ijms-22-12986-f008:**
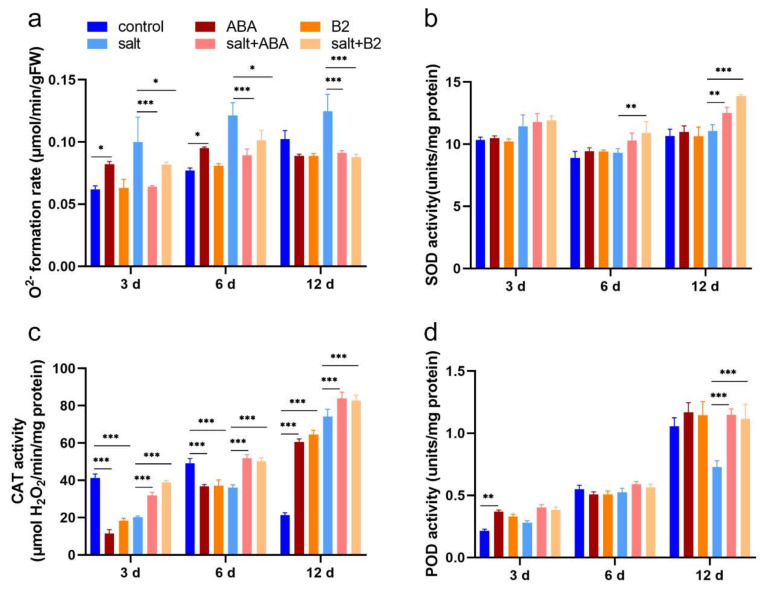
Formation rate of O^2−^ and activity of three major enzymes in the antioxidant system in the maize leaves. (**a**) O^2−^ formation rate, (**b**) SOD activity, (**c**) CAT activity, (**d**) POD activity. The concentration of B2 was 1.0 µM, and ABA concentration was 100 μM. Values were significantly different from controls by *t*-test at * *p* < 0.05, ** *p* ≤ 0.01, or *** *p* ≤ 0.001. Bars mean SD (*n* = 9).

**Figure 9 ijms-22-12986-f009:**
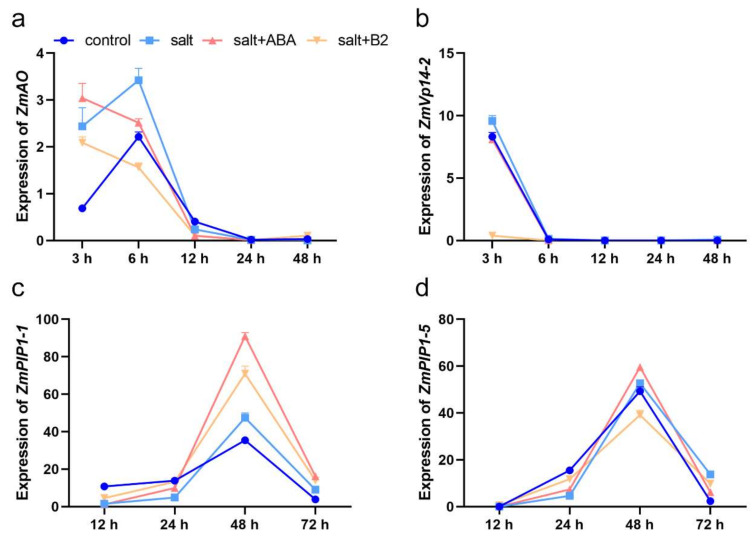
The expression levels of aquaporin and ABA biosynthesis genes. ABA was 100 μM and B2 concentration was 1μM. (**a**) The expression level of *ZmAO* gene. (**b**) The expression level of *ZmVP14-2* gene. (**c**) The expression level of *ZmPIP1:1* gene. (**d**) The expression level of *ZmPIP1:5* gene.

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
