# Peer review of "An ABA Functional Analogue B2 Enhanced Salt Tolerance by Inducing the Root Elongation and Reducing Peroxidation Damage in Maize Seedlings"

_ijms, 2021, doi:10.3390/ijms222312986_

Round 1
Reviewer 1 Report
The manuscript is interesting and wel-written. I have some points that I would like to consider them in the revised manuscript.
-in line 19 and 72, please change salt resistance to salt tolerance
-sentence, As well known, the hormonal regulation is very important in the stress response' please add reference
- the role of ABA and hormones cross talk in other crops like barley is well know, please use this to improve and extend the introduction.
- the first use of the apriviation ABA in line 49 shoud be 'the Abscisic acid (ABA)' . please correct it and remove from line 81 'Abscisic acid'. please check all apriviations and do the same.
-please change ' a little lower ' in line 88 to the exact difference in per cent, like ' this effect was .....% less than the group of...
- please add the ABA used concentration (100 µM ) either in the Figs lugend or in the Figs Key
- In Fig 3 please change ( cm) to (cm)
- in all figs please add to th legand the p≤..... to diffrentiate between~(*,** and ***) which p≤... you use for eatch
- please change the 1 μM to 1.0 μM in all MS
- Please correct the references according to the journal style. in the MS there are different style like Ref 7, 9, 22, 37 .... and others
Author Response
Dear Reviewer:
Thanks for your comments. The explanations of what changes we have made in response to reviewers’ comments are given point by point in a separate sheet of “Response to Reviewer 1 Comments”
Please see the attachment.
Sincerely,
Yuyi

Reviewer 2 Report
I suggest the authors should address all the comments in the attached pdf documents.

Author Response
Dear Reviewer:
Thanks for your comments. The changes we have made in “Revised ijms-1476355”
Please see the attachment.
Sincerely,
Yuyi
